# Oncolytic H-1 Parvovirus Enters Cancer Cells through Clathrin-Mediated Endocytosis

**DOI:** 10.3390/v12101199

**Published:** 2020-10-21

**Authors:** Tiago Ferreira, Amit Kulkarni, Clemens Bretscher, Karsten Richter, Marcelo Ehrlich, Antonio Marchini

**Affiliations:** 1Laboratory of Oncolytic Virus Immuno-Therapeutics, German Cancer Research Centre, Im Neuenheimer Feld 242, 69120 Heidelberg, Germany; t.ferreira@dkfz.de (T.F.); c.bretscher@dkfz.de (C.B.); 2Laboratory of Oncolytic Virus Immuno-Therapeutics, Luxembourg Institute of Health, 84 Val Fleuri, L-1526 Luxembourg, Luxembourg; amit.kulkarni@lih.lu; 3Core Facility Electron Microscopy, German Cancer Research Centre, Im Neuenheimer Feld 280, 69120 Heidelberg, Germany; k.richter@dkfz-heidelberg.de; 4Laboratory of Signal Transduction and Membrane Biology, The Shumins School for Biomedicine and Cancer Research, George S. Wise Faculty of Life Sciences, Tel Aviv University, 69978 Tel Aviv, Israel; marceloe@post.tau.ac.il

**Keywords:** oncolytic viruses, rodent protoparvovirus H-1PV, virus entry, clathrin-mediated endocytosis

## Abstract

H-1 protoparvovirus (H-1PV) is a self-propagating virus that is non-pathogenic in humans and has oncolytic and oncosuppressive activities. H-1PV is the first member of the *Parvoviridae* family to undergo clinical testing as an anticancer agent. Results from clinical trials in patients with glioblastoma or pancreatic carcinoma show that virus treatment is safe, well-tolerated and associated with first signs of efficacy. Characterisation of the H-1PV life cycle may help to improve its efficacy and clinical outcome. In this study, we investigated the entry route of H-1PV in cervical carcinoma HeLa and glioma NCH125 cell lines. Using electron and confocal microscopy, we detected H-1PV particles within clathrin-coated pits and vesicles, providing evidence that the virus uses clathrin-mediated endocytosis for cell entry. In agreement with these results, we found that blocking clathrin-mediated endocytosis using specific inhibitors or small interfering RNA-mediated knockdown of its key regulator, AP2M1, markedly reduced H-1PV entry. By contrast, we found no evidence of viral entry through caveolae-mediated endocytosis. We also show that H-1PV entry is dependent on dynamin, while viral trafficking occurs from early to late endosomes, with acidic pH necessary for a productive infection. This is the first study that characterises the cell entry pathways of oncolytic H-1PV.

## 1. Introduction

The rodent H-1 protoparvovirus (H-1PV) belongs to the *Parvoviridae* family, genus *Protoparvovirus* [1]. This genus also includes *Rodent protoparvovirus 1* (H-1PV, Kilham rat virus, LuIII virus, minute virus of mice (MVM), mouse parvovirus, tumour virus X, rat minute virus), *Rodent protoparvovirus 2* (rat parvovirus 1), *Carnivore protoparvovirus 1* (canine parvovirus (CPV) and feline panleukopenia parvovirus (FPV)), *Primate protoparvovirus 1* (bufavirus) and *Ungulate parvovirus 1* (porcine parvovirus (PPV)) [2,3]. Protoparvoviruses (PtPVs) are single-stranded DNA viruses with an icosahedral capsid of about 25 nm diameter. Their genomes encompass the non-structural (NS) and the viral particle (VP) transcriptional units, whose expressions are regulated by the P4 and P38 promoters, respectively. The NS transcriptional unit encodes the NS1 and NS2 proteins, whereas the VP transcriptional unit encodes the VP1 and VP2 capsid proteins and the small alternatively translated protein [4].

Owing to their ability to specifically infect, replicate and kill human cancer cells, rodent PtPVs are under investigation as potential anticancer therapies. Pre-clinical studies have revealed that H-1PV in particular has remarkable oncolytic and oncosuppressive activity in a number of cell culture and animal models of cancers from different origins [5]. Notably, H-1PV-induced cancer cell death and lysis are immunogenic and stimulate the immune system to participate in the elimination of cancer cells [6]. NS1 is the major effector of H-1PV oncotoxicity [7].

Although viral oncolytic activity is shared between rodent PtPVs, H-1PV is the only member of the genus to have reached the clinic as an anticancer therapy. In a phase I/IIa clinical trial in patients with recurrent glioblastoma (ParvOryx01), H-1PV treatment was safe, well-tolerated and associated with first evidence of anticancer efficacy. This evidence included the ability of H-1PV to cross the blood–brain barrier after intravenous administration, its wide distribution in the tumour bed, the induction of tumour necrosis and immuno-conversion of the tumour microenvironment. As a result, virus treatment led to an improved progression-free survival and median overall survival of patients in comparison with historical controls [8]. A dose-escalation phase I/IIa pilot study in patients with metastatic pancreatic cancer recently confirmed the excellent safety and tolerability of H-1PV treatment. In accordance with the results of ParvOryx01, patients who responded to the treatment showed evident changes in the tumour microenvironment and induction of specific immune responses [9].

The PtPV life cycle is strictly dependent on host cellular factors for a productive infection, from cell surface attachment and entry to virus DNA replication, gene expression, multiplication and egress. Some of these factors are frequently overexpressed or dysregulated in cancer cells. The list includes cell cycle regulators, transcription factors, modulators of the DNA damage response, kinases and cytoskeleton components (reviewed in Reference [10]). However, unlike for other PtPVs, the early steps of H-1PV infection remain to be characterised.

The first interaction between PtPVs and the target cell occurs through binding to a specific surface receptor exposed on the host plasma membrane. Cellular receptors for some PtPVs have been described, such as the transferrin receptor for CPV and FPV. H-1PV, like MVM and PPV, uses sialic acid (SA) for cell surface attachment and entry. However, it is unclear whether SA itself acts as a functional viral receptor for the virus or is a component of an as yet unidentified receptor(s) or receptor complex [3,11,12].

After docking to the cellular membrane, viruses are internalised through different pathways [13]. Clathrin- and caveolae-mediated endocytosis are two dynamin-dependent pathways, whereas macropinocytosis, lipid-raft-mediated endocytosis and caveolae/clathrin-independent endocytosis are dynamin-independent pathways [14,15]. Clathrin-mediated endocytosis is the pathway commonly used by small viruses, including PtPVs [16,17,18,19,20]. The mechanism begins with the recruitment of adaptor protein 2 (AP-2) complexes on the plasma membrane, followed by the assembly of a three-dimensional clathrin coat that leads to a progressive invagination of the membrane. Dynamin self-assembles around the vesicle neck and mediates its scission, and the vesicle is subsequently released into the interior of the cell [21]. 

PtPVs also use alternative endocytic pathways. For instance, MVM prototype strain takes at least three different endocytic routes: clathrin-, caveolae- and clathrin-independent carrier-mediated endocytosis [22]. Even though endocytosis seems to be the default entry pathway for PtPVs, differences between members of the family may contribute to the tropism of these viruses.

As the PtPV is trafficked within the cellular endosome, its capsid undergoes slow structural changes. In particular, the acidic environment exposes the catalytic phospholipase 2 domain of VP1. This conformational change promotes the digestion of the endosomal membrane, resulting in the release of viral particles from the late endosome to the cytosol [3]. Thereafter, incoming PtPV particles are transported to the nucleus in a process that is dependent on the cytoskeleton and associated motor proteins [23].

In this study, we used electron microscopy (EM) and immunofluorescence (IF), together with a number of chemical inhibitors and siRNA-mediated knockdown, to identify which of these pathways H-1PV uses to enter cancer cells. We found that H-1PV cell uptake occurs preferentially through clathrin-mediated, but not caveolae-mediated, endocytosis in cervical carcinoma HeLa and glioma NCH125 cell lines. Entry was also dependent on dynamin activity. We show that after its internalisation, H-1PV, like other PtPVs, passes through early endosomes to late endosomes/lysosomes during its cytosolic trafficking. Productive infection relies heavily on the acidic pH in the endosomes.

## 2. Materials and Methods 

### 2.1. Cells and Viruses 

The cervical carcinoma-derived HeLa [7] and the glioblastoma-derived NCH125 cell lines [24] were cultured in Dulbecco’s modified Eagle’s medium (DMEM) supplemented with 10% FBS, 100 units/mL penicillin, 100 μg/mL streptomycin and 2 mM L-glutamine (all from Gibco, Thermo Fischer Scientific, Darmstadt, Germany) in a humidified incubator at 37 °C. Both cell lines were tested for mycoplasma contamination by PCR in a regular base.

Both wild-type H-1PV and recombinant H-1PV harbouring the green fluorescent protein-encoding gene (recH-1PV-EGFP) were produced, purified and titrated as previously described [25,26].

### 2.2. Electron Microscopy

HeLa cells were seeded on punched sheets of ACLAR-Fluoropolymer films (Electron Microscopy Sciences) at a density of 8 × 10^4^ cells/well in 24-well plates. On the following day, cells were infected with H-1PV at a multiplicity of infection (MOI) of 2000 plaque forming units (pfu) per cell in DMEM 5% FBS at 4 °C for 1 h to allow virus attachment to the cell surface and promote a synchronised infection. In order to catch the internalisation event, cells were shifted to 37 °C for 0, 5, 10, 20 and 30 min. After incubation, ACLAR-Fluoropolymer films were embedded in epoxy resin for ultrathin sectioning according to standard procedures. Briefly, chemical fixation was carried out in buffered aldehyde (4% formaldehyde, 2% glutaraldehyde, 1 mM CaCl_2_, 1 mM MgCl_2_ in 100 mM Ca-cacodylate, pH 7.2), followed by post-fixation in buffered 1% osmium tetroxide and en bloc staining in 1% uranylacetate. Following dehydration in graded steps of ethanol, the adherent cells were flat-embedded in epoxy resin (mixture of glycid ether, methylnadic anhydride and dodecenyl-succinic-anhydride; Serva). Ultrathin sections of nominal thickness 60 nm and contrast-stained with lead-citrate and uranylacetate were analysed using a Zeiss EM 910 (Carl Zeiss, Oberkochen, Germany) at 120 kV and micrographs taken using a slow scan charge-coupled device camera (TRS, Olympus, Moorenweis, Germany).

### 2.3. Co-Localisation of H-1PV and Cellular Proteins by Confocal Microscopy

HeLa cells were seeded at a density of 3.5 × 10^3^ cells/spot on spot slides and grown in 50 µL of complete cellular medium. On the following day, cells were placed on ice for 15 min and then infected with wild-type H-1PV at a MOI of 500 (pfu/cell) in a total of 70 µL of 5% FCS-containing medium. At 1 h post-infection, cells were shifted to 37 °C for varied times depending on the experiment, before being fixed with 3.7% paraformaldehyde on ice for 15 min and permeabilised with 1% Triton X-100 for 10 min. Immunostaining was carried out with the following antibodies, all used at dilution 1:500 for 1 h: mouse monoclonal anti-H-1PV capsid (a conformational antibody kindly provided by Barbara Leuchs; DKFZ Virus Production and Development Unit, Heidelberg, Germany) [27], rabbit monoclonal anti-clathrin heavy chain (D3C6; Cell Signalling Technology, Leiden, Netherlands), rabbit monoclonal anti-EEA1 (3288; Cell Signalling Technology), rabbit monoclonal anti-Rab7 (9367T; Cell Signalling Technology) and rabbit polyclonal anti-LAMP-1 (CD107a) (AB2971; Merck, Darmstadt, Germany). Anti-mouse Alexa Fluor 594 IgG (A11005; Thermo Fisher Scientific, Bleiswijk, Netherlands) or anti-rabbit Alexa Fluor 488 IgG (A11008; Thermo Fisher Scientific) were used as secondary antibodies. Nuclei were stained by 4′,6-diamidin-2-phenylindol (DAPI). Images in the green channel (H-1PV), red channel (varied cellular proteins) or blue channel (DAPI) were acquired with a confocal microscope (Leica TCS SP5 II, Wetzlar, Germany). Picture analysis was carried out using the Leica LAS X Software.

### 2.4. Treatment with Inhibitors of Endocytosis Pathways

Hypertonic sucrose (Carl Roth), 0.40 M, chlorpromazine (Sigma-Aldrich Chemie GmbH, Steinheim, Germany), 2.5 µg/mL for HeLa or 5 µg/mL for NCH125, and pitstop 2 (Sigma-Aldrich Chemie GmbH), 30 µM, were used to inhibit clathrin-mediated endocytosis. Dynole™ Series Kit containing Dynole 31–2 (active drug) and 34–2 (negative control) (ab120474; Abcam, Cambridge, UK), 5 µM, were used to inhibit dynamin activity. Bafilomycin A1 (BafA1; Cell Signalling Technology), 10 nM, and ammonium chloride (NH4CL; 1145, Merck), 25 mM, were used to prevent pH acidification. The concentrations of the aforementioned inhibitors were used at the highest doses before affecting the cellular proliferation. Nystatin (Sigma-Aldrich), 10 µg/mL, and methyl-β-cyclodextrin (MβCD; Sigma-Aldrich Chemie GmbH), 10 mM, were used to inhibit caveolae-mediated endocytosis. These concentrations were selected according to the literature [28,29] and also did not affect the proliferation of the cell lines used for the experiments. 

Briefly, HeLa cells were seeded at a density of 8 × 10^4^ cells/well in 24-well plates and, on the following day, pre-treated for 45 min with the various inhibitors. Cells were then infected with recH-1PV-EGFP at a MOI of 0.2–0.3 transduction units (TU)/cell for 4 h, washed twice with PBS and grown in culture medium for an additional 20 h. At 24 h post-infection, cells were washed once with PBS, fixed with 3.7% paraformaldehyde on ice for 15 min, permeabilised with 1% Triton X-100 for 10 min and stained with DAPI. Fluorescence images of EGFP-positive cells were acquired with a BZ-9000 fluorescence microscope (Keyence Corporation, Osaka, Japan) with 4X or 10X objective. DAPI staining was used to visualise the nuclei (cells). At least two independent experiments, each performed in duplicate, were performed for every condition tested. 

### 2.5. Cell Proliferation Assay

Cell proliferation was monitored in real time through the xCELLigence system (ACEA Biosciences Inc., San Diego, CA, USA) according to the manufacturer’s instructions. Briefly, 8 × 10^4^ HeLa or NCH125 cells per well were seeded in a 96-well E-plate (Roche) in a total volume of 100 µL of complete DMEM medium. On the following day, cells were treated with different inhibitors for 45 min, and subsequently washed with PBS. Cell proliferation was monitored every 30 min in real time over a period of 72 h. Data is expressed as “Normalised cell index”, where all curves were normalised to an arbitrary value of 1.0 at the timepoint before treatment. Average values of each experimental condition assessed in triplicate are presented with the respective standard deviation (SD).

### 2.6. siRNA-Mediated Knockdown

Cells were seeded at a density of 4 × 10^4^ cells/well in a 24-well plate and grown in 500 µL of normal growth medium. After 24 h, cells were transfected with 10 nM siRNA using Lipofectamine RNAimax (Thermo Fisher Scientific, Carlsbad, CA, USA) according to the manufacturer’s instructions. For *AP2M1*, we used the *AP2M1* ON-TARGET plus Human siRNA SMARTpool (L-008170–00-0005) and, as a negative control, the plus Non-targeting pool (D-001810–10-05) (Dharmacon, Thermo Fisher Scientific). For CAV-1, we used two Silencer Select Validated siRNAs (s2446 and s2448; Life Technologies, Paisley, Scotland) and Silencer Select Negative Control #2 siRNA (4390846, Life Technologies) as a control. After 24 h, the medium was replaced, and cells were grown for an additional 24 h to allow efficient gene silencing. The cells were then infected for 24 h with recH-1PV-EGFP at 0.2–0.3 TU/cell. Cells were then washed once with PBS and processed as described above for fluorescence microscopy. At least two independent experiments, each performed in duplicate, were performed for every condition tested.

### 2.7. Western Blotting

Cells were harvested, washed in PBS, and then lysed on ice for 30 min in RIPA buffer (10 mM Tris-HCl pH 7.5, 150 mM NaCl, 1 mM EDTA pH 8, 1% NP-40, 0.5% Na-deoxylcholate and 0.5% SDS supplemented with complete EDTA-free protease inhibitor (11697498001; Roche, Mannheim, Germany). Cellular debris was removed by centrifugation, and protein concentration in cell lysates was measured by bicinchoninic acid (BCA) assay (Thermo Fisher Scientific), according to manufacturer’s instructions. SDS-PAGE analysis was performed on 50 µg of total protein extract. After separation, proteins were transferred to Hybond-P membrane (GE Healthcare, Freiburg, Germany). Immunoblotting was carried out with the following antibodies: mouse monoclonal anti-vinculin (sc-25336; Santa Cruz Biotechnology, Heidelberg, Germany) and rabbit monoclonal CAV-1 (D46G3; Cell Signalling Technology) at 1:1000 dilution. After incubation with horseradish peroxidase conjugated secondary antibodies (Santa Cruz Biotechnology) at 1:1000 dilution, proteins were revealed with Western Blot Chemiluminescence Reagent *Plus* (Perkin Elmer Life Sciences) and exposed to Hyperfilm™ ECL radiographic films (GE Healthcare).

## 3. Results

### 3.1. Electron Microscopy Analysis Reveals H-1PV within Clathrin-Coated Pits

In order to investigate the H-1PV internalisation pathway, we performed EM analysis of HeLa cells infected with H-1PV. Infection was carried out at 4 °C for 1 h to allow virus cell surface binding. Cells were then shifted back to 37 °C for various intervals to allow virus internalisation. EM analysis of infected cells at 4 °C showed the virus at the cell surface bound to thickened regions resembling clathrin-associated plasma membrane (Figure 1A) [30]. In the first 5 min after cells were shifted back to 37 °C, the invagination of clathrin-rich regions started with H-1PV particles remaining associated with these regions (Figure 1B). From 10 to 30 min, viral particles were detected both in clathrin-coated pits (Figure 1C) or in the cytosol inside completely invaginated vesicles (we found up to nine particles in a single vesicle) (Figure 1D). Furthermore, in the course of the experiment, no virus internalisation was found in vesicles with the small flask-shaped invaginations that are characteristic of caveolae-mediated endocytosis [31], which suggests that H-1PV is internalised mainly (if not exclusively) via clathrin-mediated endocytosis.

### 3.2. H-1PV Co-Localises with Clathrin Upon Entry

The EM analysis provided first evidence that H-1PV uses clathrin-mediated endocytosis to enter cells. To confirm these results independently, we checked for possible co-localisation of H-1PV and clathrin-heavy chain (CHC). To this end, HeLa cells were infected with H-1PV for 1 h at 4 °C and then shifted back to 37 °C. After 30 min, a fraction of internalised H-1PV was clearly detected in association with CHC by confocal microscopy (Figure 2), providing further evidence that H-1PV is internalised through a clathrin-dependent pathway.

### 3.3. H-1PV Enters Cells Preferentially via Clathrin-Mediated Endocytosis

Next, we investigated whether targeting regulators of clathrin-mediated endocytosis would affect H-1PV infection. A recombinant H-1PV expressing the EGFP reporter gene (recH-1PV-EGFP) was used for the experiments. This non-replicative parvovirus shares the same capsid of the wild type but harbours the EGFP gene under the control of the natural P38 late promoter, whose activity is regulated by NS1 viral protein [25]. Therefore, the EGFP signal directly correlates to the ability of the virus to reach the nucleus and initiate its own gene transcription.

The effects of various inhibitors on H-1PV entry were assessed in HeLa and in glioma-derived NCH125 cell lines. The latter, like HeLa, is highly permissive to H-1PV infection. Pharmacological inhibitors included hypertonic sucrose, chlorpromazine (CPZ) and pitstop 2. Hypertonic sucrose is a classical inhibitor that traps clathrin in microcages [32]. CPZ is a cationic, amphiphilic drug that induces the misassembly of clathrin lattices at the cell surface and on endosomes [33]. Pitstop 2 interferes with the binding of proteins to the N-terminal domain of clathrin [34]. The internalisation of TexasRed-labelled transferrin, a protein known to be exclusively internalised through clathrin-mediated endocytosis, was monitored to check the effectiveness of each treatment [35]. At the concentrations used, the three inhibitors blocked transferrin uptake efficiently but did not affect cell proliferation (Appendix A).

Pre-treatment with hypertonic sucrose decreased H-1PV transduction by more than 90% in HeLa and 80% in NCH125 cells compared to untreated cells (Figure 3A,B). When the compound was applied 3 h post-infection (by which time the virus is already internalised), no significant changes in H-1PV transduction were observed, indicating that the drug interferes with H-1PV transduction at the level of virus entry (Appendix A). Strong inhibition of H-1PV transduction was also achieved by pre-treating cells with CPZ (approximately 90% reduction in both cell lines) or with pitstop 2 (60% reduction in Hela and over 70% in NCH125 cells) compared to untreated cells.

The AP-2 complex is a heterotetramer that plays an essential role in clathrin-mediated endocytosis [21]. To confirm the involvement of clathrin-mediated endocytosis in H-1PV cell entry, we silenced the expression of *AP2M1*, the gene encoding subunit µ1 of AP-2 [36]. To this end, HeLa and NCH125 cells were transfected with either a siRNA pool targeting *AP2M1* or scrambled siRNA (negative control), and subsequently infected with recH-1PV-EGFP. Under conditions in which the silencing of *AP2M1* successfully reduced transferrin uptake, we observed a strong decrease in H-1PV transduction (over 60% compared to the scrambled siRNA-treated cells) in both cell lines (Figure 3C,D). Taken together, these results show that H-1PV uses clathrin-mediated endocytosis to enter HeLa and NCH125 cells.

### 3.4. H-1PV Does Not Enter Cells via Caveolae-Dependent Endocytosis

To investigate whether H-1PV uses pathways other than clathrin-mediated endocytosis to enter HeLa and NCH125 cancer cells, we inhibited clathrin-independent endocytosis using nystatin and methyl-β-cyclodextrin (MβCD). Both drugs selectively disrupt lipid rafts (e.g., cholesterol), including those required for caveolae-dependent entry [28,29,37]. Yet, neither nystatin nor MβCD decreased H-1PV transduction, providing evidence that the virus does not use this endocytic route to enter these cells (Figure 4A,B). Similar results were also obtained using the two drugs at lower or higher concentrations (0.1–100 µg/mL for nystatin, 0.1–100 μM for MβCD). We also carried out siRNA-mediated knockdown of *CAV1* (which encodes for caveolin-1) by using two different siRNAs. Knock-down of *CAV1* gene expression did not decrease H-1PV transduction activity compared with scrambled siRNA-transfected cells, but instead increased it (Figure 4C,D). Together, these results indicate that caveolae-dependent endocytosis is not involved in H-1PV entry of HeLa and NCH125 cells.

### 3.5. H-1PV Internalisation Is Dependent on Dynamin

Dynamin is a large GTPase with an essential role in cellular membrane fission for newly formed vesicles. It is therefore required for clathrin- and caveolae-mediated endocytosis but not for macropinocytosis [38]. We used the highly selective Dynole 34–2 to inhibit dynamin activity [39,40], and Dynole 31–2, its inactive form, as a negative control. At a concentration that blocked transferrin uptake, Dynole 34–2 drastically reduced virus transduction to just 8% in HeLa and 13% in NCH125 cells (Figure 5) compared to untreated cells. As expected, Dynole 31–2 did not have any significant effect on H-1PV transduction. These results demonstrate that dynamin plays an essential role in H-1PV infection.

### 3.6. H-1PV Hijacks Endosomes for Trafficking into the Cytosol and Acidic pH Is Required for Productive H-1PV Infection

The Rab family of proteins and their effectors play a key role in the formation, maintenance and trafficking of endosomes [41]. Early endosome antigen 1 (EEA1) is a Rab5 effector protein that is involved in sorting endocytic vesicles at the early endosome level [42]. Rab7 is considered to be the key regulator of late endosome trafficking. Lysosomal-associated membrane protein 1 (LAMP-1) is enriched in late endosomes and lysosomes, where, among other functions, it maintains lysosomal integrity and pH [43,44].

To provide direct evidence that H-1PV hijacks endocytosis for its intracellular trafficking, we infected HeLa cells with H-1PV for 1 h, fixed the cells, and then stained viral capsids as well as EEA1, Rab7 and LAMP-1 proteins with antibodies. Confocal microscopy analysis showed that H-1PV co-localised with EEA1, Rab7 and LAMP-1 markers during infection (Figure 6A). 

Next, we hypothesised that the acidic pH in the endocytic compartments would provide the environmental conditions required for a productive H-1PV infection [3] (as it does for other PtPV infections), possibly triggering the conformational changes necessary for uncoating and nuclear translocation. To test whether low endosomal pH is required for H-1PV infection, we pre-treated HeLa and NCH125 cells with ammonium chloride (NH_4_Cl), a lysosomotropic weak base [45], or bafilomycin A1 (BafA1), a blocker of vacuolar H^+^-ATPases [46]. Treatment with NH_4_Cl resulted in a strong decrease of virus transduction in both HeLa and NCH125 cells (39% and 30%, respectively) (Figure 6B,C), while treatment with BafA1 completely abolished H-1PV transduction (Figure 6D,E), in comparison to untreated cells. Together, these results provide evidence that H-1PV, like other PtPVs, requires acidic endosomal pH for a productive infection.

## 4. Discussion

H-1PV is a promising oncolytic virus. Early-phase clinical studies show that virus treatment is safe, well-tolerated and associated with first signs of anticancer efficacy. The present study aimed to fill a gap in knowledge of H-1PV biology regarding the pathways used by this virus to enter cancer cells. We chose two cancer cell lines as models to investigate the early steps of H-1PV infection: the cervical carcinoma-derived HeLa cell line and the glioma-derived NCH125 cell line. Previous laboratory studies have shown that these two cell lines are highly permissive to H-1PV infection and susceptible to its oncolytic activity [7,24,47]. 

In our study, for the first time, we show that H-1PV exploits clathrin-mediated endocytosis to enter cancer cells. EM analysis of H-1PV-infected HeLa cells revealed virus particles associated with clathrin-coated pits and then internalised inside clathrin-coated vesicles (Figure 1). In agreement with this finding, confocal microscopy analysis showed co-localisation of H-1PV and clathrin (Figure 2). Furthermore, pharmacological inhibition of clathrin-mediated endocytosis using hypertonic sucrose, CPZ and pitstop 2, as well as siRNA-mediated silencing of *AP2M1*, a key regulator of clathrin-mediated endocytosis, confirmed the heavy dependence of H-1PV on this pathway for successful entry into both HeLa and NCH125 cancer cells (Figure 3). 

Previous research has reported that other PtPVs use clathrin-mediated endocytosis to gain access to cells and progress the infection (reviewed in Reference [3]). Our results are therefore in agreement with the idea that clathrin-mediated endocytosis is the default entry pathway for PtPVs. However, a number of PtPVs have also been shown to hijack alternative endocytic pathways. For instance, MVMp uses at least three different endocytic routes, such as clathrin-, caveolae- and clathrin-independent carrier-mediated endocytosis [22]. Moreover, PPV uses both clathrin-dependent endocytosis and macropinocytosis [16]. FPV and CPV are taken up by cells via binding to the transferrin receptor, which is typically endocytosed by clathrin-mediated endocytosis. However, FPV may also enter cells through alternative internalisation mechanisms, as deletions or mutations in the internalisation motif of the transferrin receptor, while decreasing FPV cellular uptake, did not completely arrest viral infection [48]. Although we cannot completely rule out the possibility that H-1PV, like other PtPVs, can also use different pathways to enter HeLa and NCH125 cancer cells, our results seem to exclude caveolae-mediated endocytosis as a major entry pathway. Indeed, pre-treatment of cells with nystatin and MβCD, two inhibitors of caveolae-mediated endocytosis, did not decrease H-1PV transduction levels, suggesting that these drugs did not modify H-1PV entry. These results are in agreement with previous studies showing that bovine parvovirus (BPV) [17] and PPV [16] do not use caveolae-mediated endocytosis for their cell internalisation. Surprisingly, *CAV1* siRNA-mediated knockdown increased H-1PV transduction (Figure 4), suggesting that caveolin-1 (the protein encoded by *CAV1*) interferes with H-1PV infection. Human immunodeficiency virus (HIV) infection is restricted by caveolin-1 in various ways, e.g., at the transcriptional level by suppressing NF-kB p65 acetylation in macrophages, or by interacting with HIV viral proteins to impair viral infectivity [49,50]. Similarly, abundant caveolin-1 levels prevent Influenza A virus from infecting mouse embryo fibroblasts, which is reversed by depleting caveolin-1 [51]. Further experiments are required to find out whether and at what level caveolin-1 represents a negative modulator of H-1PV infection. Along these lines, caveolin-1 antagonists or inhibitors could offer an interesting strategy to improve H-1PV efficacy in cancer with elevated levels of caveolin-1.

Our study also provides important evidence that dynamin is involved in H-1PV entry (Figure 5). Dynamin is also required for MVMp entry in murine A9 fibroblasts, a process that occurs through both clathrin- and caveolae-mediated endocytosis [22]. However, the same study showed that dynamin is not involved in MVMp entry into mouse mammary cells transformed with polyomavirus middle T antigen, which instead occurs via clathrin-independent carrier-mediated endocytosis [22].

Another cell entry route used by viruses is macropinocytosis, generally described as a dynamin-independent process [52,53]. Among PtPVs, PPV has been reported to use this route of entry [16]. However, we showed that inhibition of dynamin almost completely abolished H-1PV infection, which makes it unlikely that H-1PV uses macropinocytosis as an alternative pathway to enter HeLa and NCH125 cancer cells.

Numerous viruses are known to hijack Rab-dependent pathways to enter cells. Of these, Rab5 and Rab7 GTPases are the key regulators of transport to early and late endosomes, while LAMP-1 is present mainly in late endosomes/lysosomes [41,54]. In the present study, we show that H-1PV particles co-localise with EEA1, an early endosome marker, and with Rab7 and LAMP-1, two late endosome markers (Figure 6A), indicating that H-1PV, like other PtPVs [3], uses endosomes for its cytosolic trafficking into the cells. However, a fraction of H-1PV particles is most likely trapped in LAMP1-positive lysosomes. This has been observed for other PtPVs such as MVM, for which sequestration in LAMP1-positive lysosomes limits the efficiency of its nuclear translocation [55], and CPV, which accumulates in perinuclear LAMP2-positive lysosomes [56]. 

Previous studies have shown that the acidic environment inside the endosomes changes in redox conditions, and acid proteases and phosphatases drive the conformational rearrangements in the catalytic phospholipase 2 domain of the VP1 protein [55,57,58]. These changes are required first for the release of PtPV particles from the late endosome to the cytosol, and then for translocation into the nucleus [23,57,59,60]. For instance, the endosomal acidic environment is required for a productive infection of B19V [20], CPV [18,61,62] and MVM [55,63], among other parvoviruses. In agreement with these observations, we also found that NH_4_Cl and BafA1 strongly hampered H-1PV transduction (Figure 6).

In summary, our study shows, for the first time, that H-1PV internalisation occurs via clathrin-mediated and dynamin-dependent endocytosis, while requiring endosomal acidification, with EEA1 and Rab7 involved in the infection process. However, we cannot rule out the possibility of H-1PV taking other pathways in other cell types. Important questions remain, namely which receptor H-1PV uses to initiate uptake and whether other co-receptors are involved. In addition, the exact mechanisms of endosomal escape and subsequent nuclear entry, and finally the site where the H-1PV genome becomes accessible for replication, need to be investigated [3]. A better understanding of the early steps of H-1PV (and, more generally, PtPV) infection is crucial not only to decipher viral tropism and inherent oncolytic properties, but also to improve the clinical potential of H-1PV in cancer virotherapy.

## Figures and Tables

**Figure 1 viruses-12-01199-f001:**
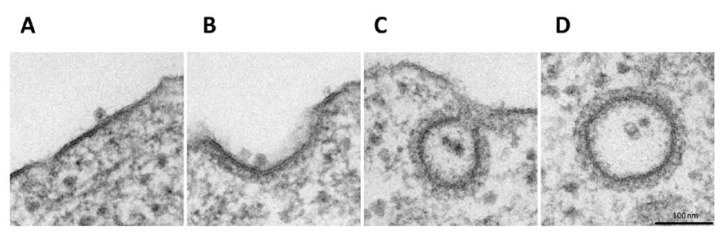
Endocytosis of H-1PV is clathrin-dependent. HeLa cells were infected with H-1PV for 1 h at 4 °C to allow H-1PV cell surface attachment but not entry. Cells were then shifted to 37 °C to allow H-1PV cell internalisation. Cells were collected every 5 min for a total of 30 min and processed for EM analysis. (**A**) At 4 °C, H-1PV particles are found attached to electro-dense (clathrin-rich) regions on the plasma membrane. (**B**) In the first 5 min after release at 37 °C, H-1PV particles are detected in early-forming clathrin-coated pits. (**C**) From 10 to 30 min, H-1PV particles moved into the cells within deeply invaginated clathrin-coated pits that were still connected to the plasma membrane, forming an hourglass-like membrane neck. (**D**) Later in the infection (10–30 min at 37 °C), H-1PV particles are seen being trafficked within the cell inside clathrin-coated vesicles.

**Figure 2 viruses-12-01199-f002:**
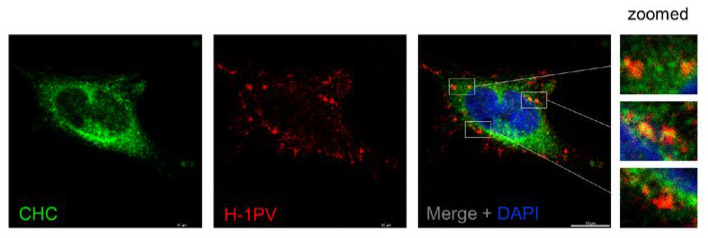
H-1PV co-localises with clathrin upon entry. HeLa cells were infected with H-1PV for 1 h at 4 °C and then shifted to 37 °C for 30 min before being processed for immunostaining using anti-H-1PV full capsid and anti-clathrin-heavy chain (CHC) antibodies. Cell nuclei were visualised by DAPI staining. Confocal microscopy analysis showed that H-1PV particles (Alexa Fluor 594, red) co-localised with CHC (Alexa Fluor 488, green) early upon infection. Three examples of regions where co-localisation is observed are framed by white boxes and shown zoomed in.

**Figure 3 viruses-12-01199-f003:**
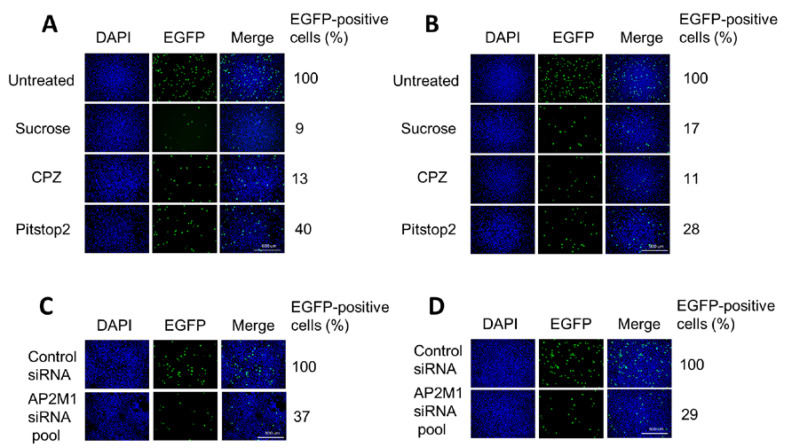
Blocking clathrin-mediated endocytosis results in a significant reduction of H-1PV transduction. (**A**) HeLa and (**B**) NCH125 cells were pre-treated with different clathrin-mediated endocytosis inhibitors (hypertonic sucrose, chlorpromazine (CPZ) or pitstop 2) or left untreated. Cells were then infected with recH-1PV-EGFP for 4 h in the presence of the inhibitor. At 20 h post-infection, cells were processed as described in the Materials and Methods (M&M) section. (**C**) HeLa and (**D**) NCH125 cells were transfected with a pool of siRNAs targeting either *AP2M1* or negative control. At 48 h post-transfection, cells were infected with recH-1PV-EGFP for 4 h and grown for an additional 20 h. Cells were then processed as described in the M&M section. Numbers represent the average percentage of EGFP-positive cells relative to the number of EGFP-positive cells observed in untreated or scrambled siRNA-transfected cells, which was arbitrarily set as 100%.

**Figure 4 viruses-12-01199-f004:**
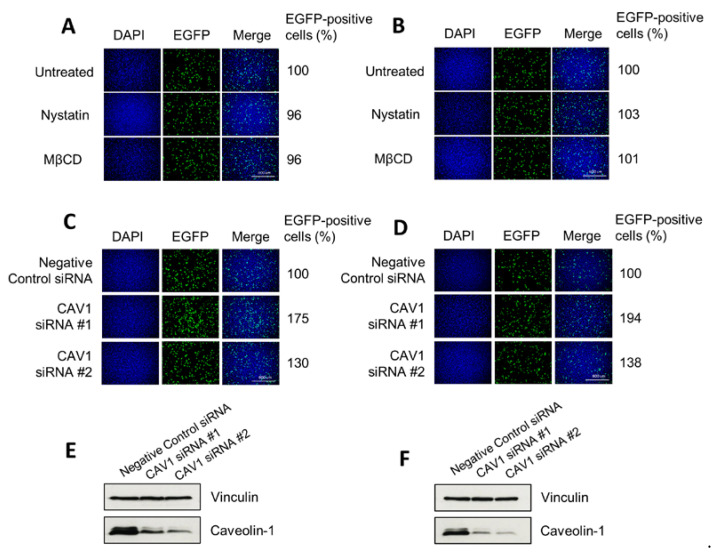
Disruption of clathrin-independent endocytosis does not decrease H-1PV transduction. (**A**) HeLa and (**B**) NCH125 cells were either pre-treated with cholesterol-sequestering drugs (nystatin or methyl-β cyclodextrin (MβCD)) for 45 min or left untreated. Cells were then infected with recH-1PV-EGFP for 4 h in the presence (or absence) of the inhibitor. At 20 h post-infection, cells were processed as described in the M&M section for immunofluorescence analysis. (**C**) HeLa and (**D**) NCH125 cells were transfected with siRNAs targeting *CAV1* or a negative control siRNA. At 48 h post-transfection, cells were infected with recH-1PV-EGFP for 4 h and grown for an additional 20 h. Cells were then processed as described in panel A. Numbers represent the average percentage of EGFP-positive cells relative to the number of EGFP-positive cells observed in untreated cells, which was arbitrarily set as 100%. The steady protein levels of caveolin-1 on lysates derived from (**E**) HeLa or (**F**) NCH125 siRNA-transfected cells were analysed by Western blotting. Vinculin was used as a loading control.

**Figure 5 viruses-12-01199-f005:**
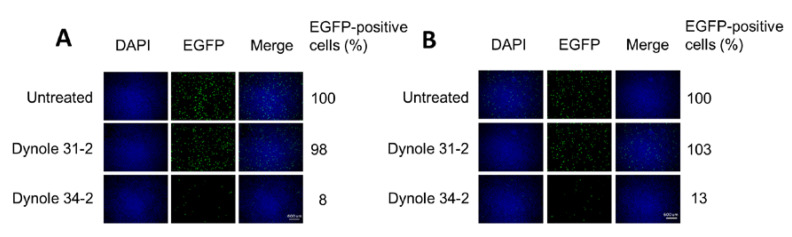
H-1PV requires dynamin activity for entry. (**A**) HeLa and (**B**) NCH125 cells were pre-treated with either Dynole 34–2 or its inactive form, Dynole 31–2. Cells were subsequently infected with recH-1PV-EGFP for 4 h in the presence of the inhibitor. At 20 h post-infection, cells were processed as described in Figure 4A. Numbers represent the average percentage of EGFP-positive cells relative to the number of EGFP-positive cells observed in untreated cells that was arbitrarily set as 100%.

**Figure 6 viruses-12-01199-f006:**
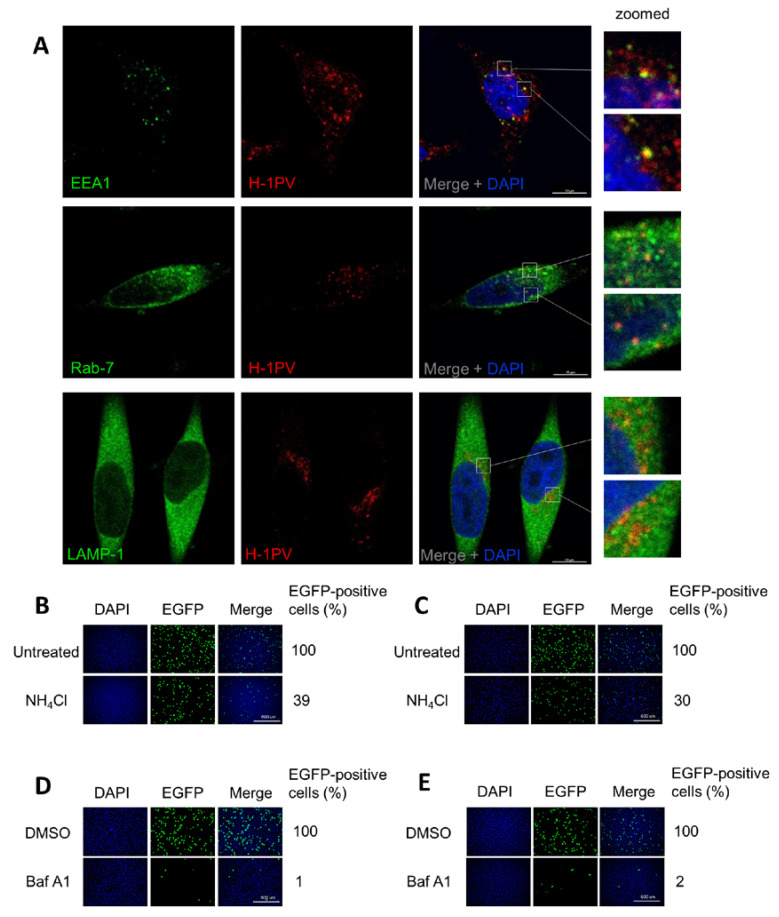
H-1PV trafficking occurs via the endosomal system with acidic pH being crucial for a productive infection. (**A**) HeLa cells were infected with H-1PV at MOI of 500 pfu/cell for 1 h at 4 °C and shifted to 37 °C for 30 min. Cells were then fixed, permeabilised and stained with DAPI and H-1PV capsid antibody, together with one of the endosomal markers (EEA1, Rab-7 or LAMP-1). Confocal microscopy analysis indicates that H-1PV particles (Alexa Fluor 594) co-localise with all three markers, EEA1, Rab-7 and LAMP-1 (Alexa Fluor 488), upon infection. Two examples of regions where co-localisation is observed are framed in white boxes and shown enlarged. (**B**,**D**) HeLa and (**C**,**E**) NCH125 cells were incubated with ammonium chloride (NH_4_CL) or bafilomycin A1 (BafA1) respectively, for 45 min, or left untreated. Cells were subsequently infected with recH-1PV-EGFP for 4 h in the presence of the inhibitor. At 20 h post-infection, cells were processed as described in the M&M section. Numbers represent the average percentage of EGFP-positive cells relative to the number of EGFP-positive cells observed in untreated cells, which was arbitrarily set as 100%.

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
