# Peer review of "Oncolytic H-1 Parvovirus Enters Cancer Cells through Clathrin-Mediated Endocytosis"

_viruses, 2020, doi:10.3390/v12101199_

Round 1
Reviewer 1 Report
This work investigates the mode of cell entry by H-1PV, a human parvovirus that specifically infects and kills cancer cells. The authors use electron microscopy to visualize virus entering cells via clathrin-dependent endocytosis. They then went on to use fluorescence microscopy to confirm that viral entry is clathrin-dependent and not dependent on other mechanisms such as caveolin. They also found that low pH assists viral infectivity, likely by viral activity to escape the endosome (activation of phosphatase of the capsid protein).
I found this well written, presented in a logical and clear way, with appropriate experiments to test their hypothesis of clathrin-mediated entry. This study advances the field of H-1PV, in particular as it relates to cell entry, which is important for its application as a cancer therapeutic.
I only found only one thing, a type, on line170,"Cell were seeded"should be "Cells were seeded".
Author Response
[Reviewer 1]
This work investigates the mode of cell entry by H-1PV, a human parvovirus that specifically infects and kills cancer cells. The authors use electron microscopy to visualize virus entering cells via clathrin-dependent endocytosis. They then went on to use fluorescence microscopy to confirm that viral entry is clathrin-dependent and not dependent on other mechanisms such as caveolin. They also found that low pH assists viral infectivity, likely by viral activity to escape the endosome (activation of phosphatase of the capsid protein).
I found this well written, presented in a logical and clear way, with appropriate experiments to test their hypothesis of clathrin-mediated entry. This study advances the field of H-1PV, in particular as it relates to cell entry, which is important for its application as a cancer therapeutic.
I only found only one thing, a type, on line170,"Cell were seeded" should be "Cells were seeded".
[Authors] - Thank you so much for the positive feedback. We are glad to hear that you found our manuscript interesting.
Reviewer 2 Report
Authors were able to support that H-1PV predominantly relies on clathrin-mediated endocytosis for cell entry.
I had just a few questions for clarification.
M&M
EM lines 117-118. An moi = 2000 seems excessive. Why did you choose to use that large an moi? Also, why is a 4 degree C incubation period necessary for PV?
Confocal lines 131-132. Similar questions for placing cells on ice then inoculating at an moi = 500.
2.4 line 154. How does TU translate to pfu when using the viral gfp construct?
2.5 line 167. Please include the calculation used to determine "normalized cell index" (& check spelling).
Westerns line 187. Why did you use a range of loading protein concentrations?
Please state why the specific concentrations of blockers were used. Did you establish what the highest dose is in the two cell lines before altering proliferation of the cells? Or did you use the lowest dose that blocked controls? Or did you optimize the dose to show the most significant results for H-1PV infection?
Results 3.3 line 263. Change Figure 2C, D to Figure 3C, D.
Author Response
Authors were able to support that H-1PV predominantly relies on clathrin-mediated endocytosis for cell entry.
[Authors] - Thank you very much for the positive feedback.
M&M
EM lines 117-118. An moi = 2000 seems excessive. Why did you choose to use that large an moi? Also, why is a 4 degree C incubation period necessary for PV?
Confocal lines 131-132. Similar questions for placing cells on ice then inoculating at an moi = 500.
[Authors] - For both the electron and confocal microscopy analysis, we incubated cells at 4 °C to allow virus attachment while preventing it from entering the cells. This way, we could perform a synchronised infection. This information has been added in the M&M lines 119-120.
In the electron microscopy analysis, the high multiplicity of infection (MOI) can be explained by the fact that ultrathin sections of infected cells need a high number of viruses for a successful detection.
2.4 line 154. How does TU translate to pfu when using the viral gfp construct?
[Authors] - The recombinant H-1PV expressing the EGFP reporter gene (recH-1PV-EGFP) is a non-replicative virus (it replicates only if the VP gene unit is provided in trans by mean of a helper plasmid), and therefore, titrated in each cell line by calculating the percentage of EGFP-positive cells and expressed as TU/mL. On the other hand, the wild-type H-1PV was titrated using a standard plaque assay in NB324K cells and titre was expressed as pfu/mL. It is not really possible to convert TU of a recH-1PV-EGFP in pfu as this virus will not form plaques.
2.5 line 167. Please include the calculation used to determine "normalized cell index" (& check spelling).
[Authors] - The calculation used has been included in the M&M lines 177-178.
Westerns line 187. Why did you use a range of loading protein concentrations?
[Authors] - Now corrected. Amount of protein loaded was 50 μg (line 199).
Please state why the specific concentrations of blockers were used. Did you establish what the highest dose is in the two cell lines before altering proliferation of the cells? Or did you use the lowest dose that blocked controls? Or did you optimize the dose to show the most significant results for H-1PV infection?
[Authors] - The inhibitors were used at the highest dose that did not affect cellular proliferation. For the inhibitors targeting clathrin-mediated endocytosis, a functional assay (transferrin uptake which only occurs via clathrin-mediated endocytosis) was also performed to check whether at the concentrations used the inhibitors were effective in blocking the pathway. This information has been now provided in the M&M section lines 153-154. Concentrations of nystatin and methyl cyclodextrin were selected from the literature. This information has been added at line 156 and corresponding references cited (new references N. 27 and 28). Nevertheless, we tested lower and higher doses of both drugs, and results were equivalent (line 294-295).
Results 3.3 line 263. Change Figure 2C, D to Figure 3C, D.
[Authors] - Corrected. Thank you so much.